# Antioxidant and Antimicrobial Properties of *Helichrysum italicum* (Roth) G. Don Hydrosol

**DOI:** 10.3390/antibiotics11081017

**Published:** 2022-07-28

**Authors:** Katja Bezek, Katja Kramberger, Darja Barlič-Maganja

**Affiliations:** Faculty of Health Sciences, University of Primorska, Polje 42, SI-6310 Izola, Slovenia; katja.kramberger@fvz.upr.si (K.K.); darja.maganja@fvz.upr.si (D.B.-M.)

**Keywords:** *Helichrysum italicum*, hydrosol, antioxidant activity, cytotoxicity, antimicrobial activity

## Abstract

(1) Background: According to the emergence and spread of antibiotic resistance, there is an urge for new promising substances. The purpose of the study was to test the antioxidant, cytotoxic and antimicrobial properties of the *Helichrysum italicum* (Roth) G. Don essential oil (EO) and hydrosol. (2) Methods: The antioxidant potential was determined using the DPPH (2,2-diphenyl-1-picrylhydrazyl) method. The cytotoxicity for human skin and intestinal cells was tested using primary and immortalized cell line models. The minimum inhibitory concentration (MIC) of hydrosol was then determined for six bacterial strains covering four commonly reported food pathogens. Further on, the hydrosol at a concentration of 1/8 MIC was used to test the antiadhesive effect by the crystal violet (CV) staining method. (3) Results: the EO showed a 100-times higher antioxidant and 180- to 25.000-times higher cytotoxic activity, when compared to hydrosol. Nevertheless, all bacterial strains, with the exception of *Staphylococcus aureus*, were sensitive to hydrosol in the range of 12.5% (V/V) for *Campylobacter jejuni*, to MIC values of 100% (V/V) for *Escherichia coli* and *Pseudomonas aeruginosa*. The antiadhesive potential of hydrosol was also shown. (4) Conclusions: Even though hydrosols are a by-product of the EO distillation process, they possess valuable biological activities.

## 1. Introduction

Antimicrobial resistance (AR) is a growing global problem affecting humans, animals and the environment. Moreover, due to the absence of surveillance systems and the high heterogeneity of data collection, the exact number of cases is still hard to determine [1,2]. In terms of food production chains, the presence of AR bacteria or genes pose a serious threat for public health [3]. As presented by Caniça et al. [3], the co-circulation of commensal and pathogenic bacteria in different reservoirs enables the spreading of resistance and the rapid emergence of multidrug-resistant strains. While the lack of new antimicrobials exacerbates the problem, it is necessary to identify new agents for therapeutic strategies against AR [4].

Natural products, which are the result of the countless possible interactions among the millions of diverse terrestrial and marine species existing worldwide, are evolutionarily optimized as intrinsic drug-like molecules and play an increasingly crucial role in drug discovery owing to their great chemical diversity. The wide chemical space occupied by the secondary metabolites of living organisms and microorganisms is an authentic example of how combinatorial chemistry has been performed by nature for thousands of years, giving a real arsenal of drug lead candidates [5]. As such, plants are a primary source of several natural medicinal products that have been used since ancient times in health promotion, as well as in treatment and prevention of various diseases. Plant natural products exert anti-inflammatory [6], anticarcinogenic [7], anti-diabetic [8], antibacterial, antifungal, antiviral, antimutagenic and antiallergic properties which are often associated with their antioxidant activity [9,10,11]. One particular group of plant secondary metabolites are volatile compounds, also known as essential oils (EOs), which have been used since ancient times in many different traditional healing systems all over the world, due to their biological activities [12]. Due to their antioxidant and antimicrobial properties, these preparations have also found applications as naturally occurring antioxidant and antimicrobial agents in the field of pharmacology, phytopathology, as well as medical and clinical microbiology, for extending the shelf life of food products, or in agriculture as biopesticides [13,14,15]. EOs and their bioactivity have been intensively investigated over recent years. They are recognized to display important antioxidant properties exerting protective effects against cellular oxidative stress [16]. They are also described as potent antimicrobial agents against many foodborne bacteria such as *S. enterica*, *S. aureus* [17,18] as well as pathogenic fungi such as *Candida* spp. and *Aspergillus* spp. [19,20]. Chemically, the EOs are a complex mixture of aromatic compounds belonging to different chemical classes; from predominating terpenoids to aldehydes, ketones, ethers and esters, alcohols and hydrocarbons, that are extracted from the plant by diverse preparation methods [12]. During the steam distillation of the plant material, a small fraction of polar, oxygenated, water-soluble constituents are trapped in a condensate water stream. These side products of EO preparation, so-called hydrosols (i.e., hydrolat, aromatic or rose water), often remain unutilized despite many valuable compounds and their biological features [21]. The liquefied water phase is enriched with certain amounts of the plant’s volatile constituents that are partly or completely soluble in water. Therefore, hydrosols can be used in perfumery, cosmetics, food flavoring, aromatherapy and traditional therapies [22,23]. Since hydrosols are much more diluted solutions compared to EOs, they can be applied to the skin directly.

*H. italicum* (Roth) G. Don is a typical Mediterranean plant that can be found along the east coast and on the islands of the Adriatic Sea. Already in ancient times, this plant was valued for its pharmacological activities and used in traditional medicine due to its anti-inflammatory [24], antibacterial [25] and antioxidant [26] properties. The most frequently reported therapeutic applications are related to respiratory, digestive and skin inflammatory conditions as summarized by Viegas et al. [27]. Although a large variety of chemically different *H. italicum* extracts have been prepared and tested, the most significant pharmacological properties related to skin applications have been attributed to the EO [28]. The EO of *H. italicum* is one of the most popular essential oils in cosmetics as it stimulates the blood circulation in the skin, strongly regenerates the skin and helps to reduce the appearance of fine lines and wrinkles. Due to its scent, the EO is used as a fragrance in soaps, cosmetics and perfumes. In addition, several studies have shown that *H. italicum* EO has an antimicrobial and antifungal activity and is effective against bacteria and fungi that can cause skin irritations, infections and delay wound healing [23]. *H. italicum* EO exhibits interesting antimicrobial activity that seems to be due to the large diversity of its chemical compounds [29]. A great variability in the chemical composition of the *H. italicum* EO has been reported, which is dependent on the factors such as geographical origin, plant growth stage and environmental conditions. EOs from the Balkan are characterized by a high content of α-pinene (22%), followed by γ-curcumene (10%), β-selinene (6%), neryl acetate (6%) and β-caryophyllene (5%), while along the Adriatic Coast the main compounds are: α-curcumene (15–29%) or γ-curcumene or α-pinene (25–30%), and neryl acetate (4–14%) [30]. While the biological effects of *H. italicum* EOs are quite well characterized, there is a lack of data on hydrosol, since studies investigating *H. italicum* hydrosol are very scarce. Due to the significant increase in the commercial exploitation of the wild *H. italicum* populations, a large amount of agroindustry residues and by-products remain unused. While the potential for applications of hydrosols is poorly investigated, the utilization of this waste material is important for the conservation of the natural resources [31]. Besides the use in the cosmetics, hydrosols have shown a great potential for application in the agro-food industry to hamper the development of pathogenic and spoiling microorganisms in food products and in working environments, as well as in removing microbial biofilm [32]. This complex and highly structured community of cells, encapsulated in a self-produced extracellular matrix, enables them to withstand harsh environmental conditions [33]. Moreover, adhesion and subsequent biofilm formation are considered as factors contributing to the ability of microbes to combat antimicrobials [34]. Due to the emergence and spread of microbial drug resistance, there is a high interest in developing and testing new strategies to combat infections including targeting biofilms, medicinal plants and phytocompounds [35].

The aim of our study was to test and compare the antioxidant and cytotoxic properties of *H. italicum* (Roth) G. Don EO and hydrosol. Due to its sufficient antioxidant properties and lower cytotoxicity when compared to *H. italicum* EO, hydrosol was then used in further studies of antimicrobial and antiadhesion assays against four commonly emerging food borne pathogens.

## 2. Results and Discussion

### 2.1. Antioxidant Activity

In our study, the antioxidant activity of *H. italicum* EO and hydrosol was evaluated by 2,2-diphenyl-1-picrylhydrazyl (DPPH) radical-scavenging test. The total antioxidant activity of the EO was 684.66 ± 94.22 µg of ascorbic acid equivalents mL^−1^, whereas the antioxidant potential of the hydrosol was equivalent to 4.88 ± 0.59 µg of ascorbic acid mL^−1^. The latter is with accordance to previous studies comparing the antioxidant activity of the EO and hydrosol in other medical plants and herbs [36,37]. Although the EO showed more than a 100-times higher antioxidant activity compared to hydrosol, the antioxidant potential of the latter was evaluated as satisfactory.

The most abundant compounds of EO used in our study were the sesquiterpene hydrocarbons, with a predominance of γ-curcumene, β-selinene and trans-β-caryophyllene, as well as monoterpenes, with neryl acetate and α-pinene as two major components (Appendix A). The chemical composition of the EO is in accordance with previously reported literature data showing γ-curcumene as a major component reported in *H. italicum* EO from Serbia [38]. It has also been shown that the components dominating the EO isolated from *H. italicum* plants collected along the Adriatic coast were α-pinene and sesquiterpene hydrocarbons (α- and γ-curcumene), as well as neryl acetate [39]. Importantly, the predominant concentration of γ-curcumene in the used EO could indicate the high antioxidant potential. As shown before, an EO of similar composition can be used as a flavoring agent and a natural antioxidant in preventing deterioration of foodstuff, beverage products and pharmaceuticals [40]. Nevertheless, the chemical composition of the EO demonstrates intraspecific differences in response to environmental factors (e.g., geographic origin, vegetation cycle, soil properties, altitude and climatic conditions) and can significantly influence their biological activity [38]. The main limitation of the present study is the lack of data about chemical composition of the hydrosol used. However, to our best knowledge there is no open data available about the chemical composition of *H. italicum* hydrosols with the exception of a recent publication by Ferraz et al. [41]. The authors showed the significant difference in chemical composition between EO and hydrosol which was mainly composed of α-terpineol (30.5%), followed by carvacrol (29.6%) and 1,8-cineole (15.4%), whereas EO contained γ-curcumene (16.0%), italicene (12.5%), neryl acetate (11.5%), α-curcumene (10.1%) and α-pinene (7.4%) [41], which is very similar to the EO used in our study. Due to our study results and the observed difference in antioxidant activity, the biological properties of the tested *H. italicum* EO and hydrosol could be attributed to the variable chemical composition. An analysis of the chemical profile of two types of products derived from the same plant is crucial, as emphasized earlier.

### 2.2. Cytotoxicity

*H. italicum* EO and hydrosol also showed different levels of cytotoxicity when applied to individual cell lines of skin or intestinal origin. The cytotoxicity of the EO was higher when compared to hydrosol for all cell lines; 180-times higher for immortalized cell lines Caco-2 and A375, approx. 900-times for primary epidermal melanocytes (PEM) and 25.000-times higher for CCD112CoN primary colon fibroblasts. The maximal nontoxic concentrations (MNCs) of the EO and hydrosol for the tested cell lines are listed in Table 1.

Furthermore, at the tested concentration of 0.084% (V/V) EO, the highest sensitivity was shown for PEM with no viability detected, followed by Caco-2 (approx. 30% viability) and A375 (approx. 75% viability), while CCD112CoN cells were not tested at this concentration due to the higher sensitivity shown already at lower concentrations. This highlights a potential use of *H. italicum* EO as an anticancer agent for the prevention and treatment of diseases which is in accordance with the shown induction of apoptosis in MCF-7 and HeLa cells [39]. However, the cytotoxicity at lower concentrations for primary cell lines can still be a limiting factor of EO utilization. For instance, the 0.5% V/V of EO in gel and ointment was shown to be suitable as a topical herbal remedy for improved healing wounds in a diabetes rat model [42], while increasing the dose is not recommended due to the potential for skin irritation [43].

On the contrary, hydrosol showed no cytotoxic effects in the maximal tested concentration for CCD112CoN cells (25% (V/V)) or Caco-2 cells (50% (V/V)). Furthermore, 25% (V/V) of hydrosol significantly affected the viability of A375 cells but not the PEM, for which minimum cytotoxic concentration was 50% (V/V) (approx. 55% viability). Given that hydrosol is used primarily in cosmetics for external use, its stronger cytotoxic effect towards malignant melanoma epithelial cells, when compared to primary epidermal melanocytes, indicates a potential use of hydrosol to influence the early stages of malignant melanoma epithelial cells development without the side effect for healthy epidermal melanocytes. The observed difference in cytotoxicity of the tested *H. italicum* EO and hydrosol could be attributed to the variable chemical composition between the two, as well as to the fact that hydrosols are primarily more diluted solutions in comparison to EOs.

### 2.3. Minimal Inhibitory Concentrations

As hydrosol proved sufficiently active in terms of antioxidant activity and showed lower cytotoxicity when compared to *H. italicum* EO, it was used in further studies of antimicrobial and antibiofilm effects. The minimal inhibitory concentrations of *H. italicum* hydrosol are shown in Table 2. All tested bacterial strains, with the exception of *S. aureus* 25923, were susceptible. The highest antimicrobial effect was shown in the case of *C. jejuni* strains (12.5% (V/V)), followed by *E. coli* and *P. aeruginosa,* both inhibited by 100% V/V and *S. aureus* showing resistance to the undiluted *H. italicum* hydrosol.

In recent years, hydrosols have emerged as potential antimicrobial candidates to control pathogenic and spoilage microorganisms [44]. As reviewed by D’Amato et al. [32], the application of hydrosols is promising in food products and in working environments, as well as in removing microbial biofilm [32]. In our study, a high antimicrobial potential of *H. italicum* hydrosol against *C. jejuni* was shown (Table 2) as well as against both clinical isolates that possess ciprofloxacin and tetracycline resistance. While *E. coli* and *P. aeruginosa* were inhibited by the maximal concentration used, there was no antimicrobial effect observed in the case of *S. aureus*, as shown before for other plant hydrosols [45]. However, for the EO of *H. italicum* from central Europe, antimicrobial activity towards *S. aureus* has been reported [46]. Comparing antibacterial activity of EOs and deodorized extracts of *H. italicum*, higher antibacterial activity against *P*. *aeruginosa*, *C*. *albicans*, *S*. *aureus* and *E*. *coli* was noticed for deodorized extracts (crude dry methanol and ethyl acetate extracts) [47]. Observed differences in the bioactivity of *H. italicum* hydrosol with those from the literature regarding the *H. italicum* EO are expected, since there were differences in plant material and extraction procedures used, which are the main factors affecting the chemical composition of the samples.

Although not tested in the present study, *H. italicum* EO has been extensively studied for its antibacterial and antifungal activities [29,48]. Its antimicrobial activity can be attributed to the monoterpene or sesquiterpene compounds, or more likely to the synergistic action of all bioactive components present in the EO as well as in the hydrosol of *H. italicum*. Among them, neryl acetate and α-pinene were proven to have a strong antibacterial activity [49,50]. In previous studies, EOs from various *H. italicum* subspecies were tested against different bacteria belonging to Gram-positive and Gram-negative species [51,52]. Rossi et al. [53] demonstrated that the EO obtained from endemic plants of Corsica was more effective against the Gram-positive bacterium *S. aureus* than against the Gram-negative strains *E. coli*, *Enterobacter aerogenes* and *P. aeruginosa* [53]. It is known that the antimicrobial mechanism of the EO depends not only on its chemical composition but also on the interactions between its compounds that can be synergistic or antagonistic. Therefore, the effect could be strain-specific and different when assessed on planktonic or sessile cells [52]. Moreover, the inconsistency in results when comparing the biological properties such as antioxidant and antimicrobial potential of the *H. italicum* EO could be attributed to the different chemical composition as shown before [54].

### 2.4. Antiadhesive Effect

The antiadhesive properties of the *H. italicum* hydrosol in subinhibitory concentrations (1/8 MIC) were tested after a 24 h incubation time using crystal violet assay (Figure 1). The results showed a statistically significant decrease in the optical density values of *E. coli* (*p* < 0.001), *P. aeruginosa* (*p* = 0.019) and *S. aureus* (*p* = 0.024) when compared to positive controls. There was no antiadhesive effect observed in the case of *C. jejuni* strains.

Bacterial strains used in this study are representatives of commonly emerging food borne pathogens known for their persistence in food-processing environments due to the biofilm formation. Moreover, *S. aureus*, *E. coli* and *P. aeruginosa* stand as the leading cause of hospital-acquired infections worldwide [55], often possessing the resistance to clinically important antibiotics [56]. In the literature, there is an increasing number of studies exploring the antimicrobial and antibiofilm properties of herbs, spices and their derivatives, such as EOs and other extracts [35,57]. Although the studies mainly focus on the primary products of the process, e.g., EOs, the distillation waste which usually remains unutilized still contains many valuable compounds in the plant material and water residues. Hydrosols may show an even higher antimicrobial and antibiofilm activity when compared to commonly used industrial biocides, such as benzalkonium chloride [58]. For example, the antibiofilm effect of hydrosols on Gram-positive and Gram-negative bacteria in monoculture as well as in mixed-culture biofilm was shown before [58,59,60]. Our results show a difference in the efficacy of hydrosol on microbial biofilms regarding bacterial species (Figure 1). Plant- and microorganism-derived compounds such as EOs, antimicrobial peptides, polyphenolic extracts, algae extracts, probiotic derived factors, D-amino acids and glycolipid biosurfactants showed a potential to control *Campylobacter* biofilms [61]. Although antibacterial action of hydrosol was observed for *C. jejuni* strains (Table 2), there was no antibiofilm effect. That could be due to the very low concentration of hydrosol used (1.6% V/V), thus affecting the bioactive compounds’ availability for antibiofilm action. On the other hand, a significant antibiofilm effect was observed for the other three strains tested (Figure 1). As reported before, plant hydrosols reduced the levels on *E. coli* O157:H7 loads of fresh cut apple and carrot samples [62]. Furthermore, in the case of *S. aureus* and *P. aeruginosa*, compounds of EOs and hydrosols were effective in preventing biofilm formation and also in eliminating the formed biofilm [45,60]. However, despite reports of a higher biofilm inhibition activity for EOs when compared to hydrosols [45], their practical application may still be hampered by some limitations, such as strong smell and taste and difficulties in rinsing from surfaces [32]. In this context, hydrosols meet the requirements to be used as natural antimicrobials in the food industry and sanitizing solutions for working surfaces, as well as a valuable alternative for the prevention of biofilm formation, thereby limiting the emergence of antimicrobial resistance.

## 3. Materials and Methods

### 3.1. Tested Material

The EO and hydrosol used in the present study were purchased from Aromara (Aromara d.o.o., Harmica, Croatia) with a product certificate provided. The EO was prepared from flowers and shoots of *H. italicum* (Roth) G. Don plant grown in Croatia, by water steam distillation using Clevenger-type apparatus as described before [63]. The list of chemical constituents of *H. italicum* EO series used in this study was provided by the company and is listed in the Appendix A. Briefly, among the most abundant compounds in the sample were γ-curcumene (18.1%), neryl acetate (10.8%), β-selinene (9.9%), trans-β-caryophyllene (8.4%), α-selinene (6.9%), ar-curcumene (5.8%) and α-pinene (5.6%). According to the manufacturer’s policy, the chemical analysis of hydrosol is not carried out routinely for each series and could not be provided.

### 3.2. Antioxidant Assay

The antioxidant activity of *H. italicum* EO and hydrosol was measured in terms of hydrogen-donating or radical-scavenging ability in the 1,1-diphenyl-2-picrylhydrazyl (DPPH) radical assay. The assay was performed as reported previously by Zegura et al. [64], with minor modifications. Briefly, reaction mixtures containing 50 or 0.2% (V/V) of the hydrosol or EO sample, respectively, and 0.1 mM DPPH in methanol were incubated in the dark at ambient temperature for 60 min in 96-well microtiter plate. The decrease in absorbance of the free radical DPPH was measured at 515 nm using a microplate reader Infinite F200 (Tecan Group Ltd., Zurich, Switzerland). Ascorbic acid was used as positive control. The free radical scavenging activity was calculated as the percentage of DPPH radical that was scavenged and expressed as mg of ascorbic acid equivalents per ml of plant sample. Two independent experiments with 2 replicates each were performed, respectively.

### 3.3. Cytotoxicity

#### 3.3.1. Cell Models

Following mammalian cell lines were used as the test models: epithelial cells of malignant melanoma A375 (ATCC^®^ CRL1619™), epithelial cells of colorectal adenocarcinoma Caco-2 (ATCC^®^ HTB37™), primary colon fibroblasts CCD112CoN (ATCC^®^ CRL1541™) and primary epidermal melanocytes (PEM) (ATCC^®^ PCS200013™). The cell lines used were purchased from ATCC^®^ (Manassas, VA, US). All the cells, except primary melanocytes, were grown in Dulbecco’s modified Eagle’s Medium (DMEM) (Sigma, St. Louis, MO, US), supplemented with 10% fetal bovine serum (FBS) and 2 mM glutamine at 37 °C and 5% CO_2_. Primary melanocytes were cultured in Dermal Cell Basal Medium (ATCC^®^ PCS200030) supplemented with Adult Melanocyte Growth Kit (ATCC^®^ PCS200042).

#### 3.3.2. Cytotoxicity Assay

The cytotoxicity of *H. italicum* products was determined using PrestoBlue™ Cell Viability Reagent (Life Technologies, Darmstadt, Germany). Cells were seeded onto 96-well plates at a density of 5–10 × 10^3^ cells/well and incubated overnight at 37 °C to attach. The medium was then replaced with a fresh complete medium containing 0.01–2 mg/mL (0.001–0.225% V/V) of EO (pre dissolved in DMSO) or 0.5–50% V/V of hydrosol and incubated for 24 h (37 °C; 5% CO_2_). After the incubation period, 10 µL of the PrestoBlue™ reagent was added directly to the cells. After 30 min, the fluorescence signal was measured (λ_EX_ = 535 nm, λ_EM_ = 595 nm) and the cell viability determined by comparing the response of the cells treated with *H. italicum* products to that of the vehicle (up to 1% DMSO or water) treated cells. Five individual wells were measured per treatment point.

### 3.4. Antimicrobial and Antibiofilm Activity

#### 3.4.1. Bacterial Strains and Growth Conditions

To test the antimicrobial and antiadhesive properties of *H. italicum* hydrosol, four common bacteria species were used: *Campylobacter jejuni* 81–176, *C. jejuni* 3B-432 and 3B-425 (clinical isolates, NLZOH, Slovenia), *Escherichia coli* ATCC 25922, *Pseudomonas aeruginosa* ATCC 27583 and *Staphylococcus aureus* ATCC 25923. The strains were maintained at −80°C in Brain Heart Infusion broth (BHI, Sigma-Aldrich, Darmstadt, Germany) supplemented with 20% glycerol (Sigma-Aldrich, Darmstadt, Germany). Prior to experiments, the strains were subcultured on Müller-Hinton Agar (MHA, Sigma-Aldrich, Darmstadt, Germany) aerobically at 37 °C and *C. jejuni* strains at 42 °C under micro-aerobic conditions (85% N_2_, 5% O_2_, 10% CO_2_). For inoculum preparation 5 mL of Müller-Hinton Broth (MHB, Sigma-Aldrich, Darmstadt, Germany) was inoculated with one bacteria colony and incubated overnight at the same conditions as subcultivation. The overnight cultures were then diluted in the ratio 1:100 to the final bacterial concentration in the experiment that was approximately 1 × 10^7^ CFU/mL.

#### 3.4.2. The Antimicrobial Susceptibility Assay

To determine the antimicrobial activity of *H. italicum* hydrosol, the minimal inhibition concentration (MIC) assay was used. Briefly, microdilution susceptibility testing was performed in flat-bottom 96-well clear plates (TPP Techno Plastic Products AG, Trasadingen, Switzerland) containing MHB medium (100 μL) in each but not in the first column. After that, 100 μL of hydrosol was added to first and second column followed by the two-fold serial dilution across the plate. Each well was then inoculated with 20 μL of prepared bacterial culture. Plates were incubated aerobically at 37 °C and at 42 °C under microaerobic conditions for *C. jejuni* strains. Bacterial viability was determined using PrestoBlue™ Cell Viability Reagent (Life Technologies, Darmstadt, Germany). The fluorescence signal was read using a microplate reader (Infinite f200, Tecan Trading AG, Männedorf, Switzerland). The MICs were defined as the lowest concentration of the hydrosol where no metabolic activity was detected after 24 h incubation time. All of the MIC measurements were carried out in three technical and two biological replicates. The control wells were prepared with culture medium, with the bacterial suspension only, or alternatively with the hydrosol dilutions only.

#### 3.4.3. The Antiadhesion Assay

The adhesion ability was quantified using crystal violet (CV) colorimetric assay, as described previously [65]. Briefly, for each strain the suspension of hydrosol dilutions and overnight bacterial culture was prepared. Eight wells of a 96-well polystyrene flat bottom microtiter plate (TPP Techno Plastic Products AG, Trasadingen, Switzerland) were then inoculated by 200 µL of prepared suspension and incubated for 24 h at 37 °C in aerobic conditions and for the *C. jejuni* strains in microaerobic conditions. The positive control contained overnight bacterial culture diluted in sterile MHB and the negative controls contained hydrosol dilutions without bacterial culture. After incubation, the suspensions were aspirated and the wells were washed twice with sterile phosphate buffered saline (PBS; Oxoid, Hampshire, UK). Plates were then dried at 60 °C for 10 min and stained adding 200 μL of 1% CV solution (Merck KGaA, Darmstadt, Germany) for 15 min. After that, the stain was aspirated and the wells were washed under tap water and finally dried at 60 °C for 10 min. The bounded CV was then released by adding 200 μL of 96% ethanol (Sigma-Aldrich Co., St. Louis, MO., US) and the optical density of distaining dye solution was measured at 595 nm using a microtiter plate reader (Infinite f200, Tecan Trading AG, Männedorf, Switzerland).

### 3.5. Statistical Analysis

The Mann–Whitney U test was performed in SPSS software version 26 (IBM, Armonk, NY, USA) in order to confirm the statistical significance (*p* < 0.5) of differences between the crystal violet OD values of each experimental set of wells relative to the positive control. Results were expressed as mean ± standard deviation.

## 4. Conclusions

Hydrosols are the side products of the EO preparations and are obtained by the dispersion of the plant materials via hydrodistillation. The liquefied water phase is enriched with certain amounts of the plant’s volatile constituents that are partly or completely soluble in water. The results of the present study indicate that the waste products from the distillation of *H. italicum* contain valuable bioactive compounds that possess antioxidant activity. Although there was no evident cytotoxic effect of *H. italicum* hydrosol in 25% V/V tested concentration for primary fibroblasts nor epidermal melanocytes, whereas for melanoma cells there was, additional studies for the confirmation of its anticancer properties are necessary. Moreover, promising antimicrobial activity against *C. jejuni* strains, *E. coli*, *P. aeruginosa* and antibiofilm activity towards *E. coli*, *P. aeruginosa* and *S. aureus* indicate a potential for use in the food industry to conquer the problem of spoilage and pathogenic bacteria. Therefore, the *H. italicum* hydrosol as a by-product of EO production may serve as a natural antimicrobial agent that could provide alternative or additional way of biotic (e.g., skin) and abiotic surfaces disinfection.

## Figures and Tables

**Figure 1 antibiotics-11-01017-f001:**
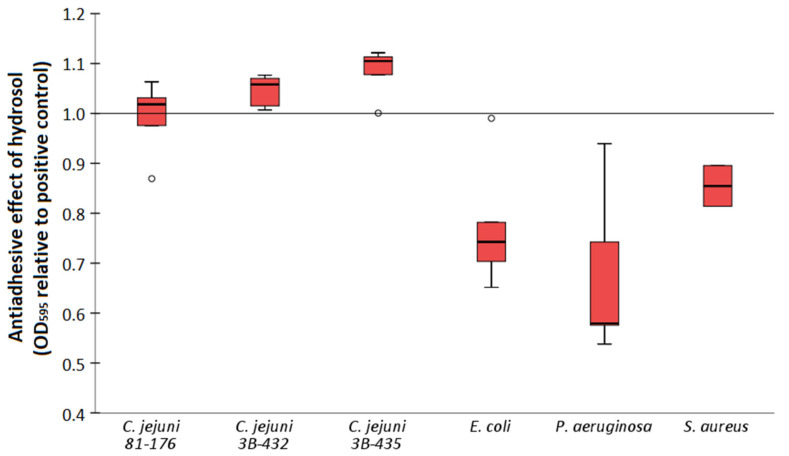
Antiadhesive properties of *H. italicum* hydrosol expressed as optical density values (OD_595_) relative to positive control.

**Table 1 antibiotics-11-01017-t001:** Maximal nontoxic concentrations (MNTs) of *H. italicum* essential oil and hydrosol for different cell lines.

Cell line	Essential Oil	Hydrosol
	MNT (% V/V)	
Caco-2	0.056	50
CCD112CoN	0.001	25
A375	0.056	10
PEM	0.028	25

MNT—maximal nontoxic concentration, PEM—primary epidermal melanocytes.

**Table 2 antibiotics-11-01017-t002:** Minimal inhibitory concentrations (MICs) of *H. italicum* hydrosol.

Bacterial Strain	MIC (% V/V)
*C. jejuni* 81-176	12.5
*C. jejuni* (3B-432)	12.5
*C. jejuni* (3B-435)	12.5
*E. coli*	100
*P. aeruginosa*	100
*S. aureus*	>100

## Data Availability

The data presented in this study are available on request from the corresponding author.

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
