# Peer review of "Antioxidant and Antimicrobial Properties of Helichrysum italicum (Roth) G. Don Hydrosol"

_antibiotics, 2022, doi:10.3390/antibiotics11081017_

Round 1

Reviewer 1 Report

The manuscript is well written and presents an exciting application for H. italicum hydrosol.  

Biological tests were well performed and results presented in a good way.

My only concern is related to the chemical composition of the samples.

Why did the authors test only one manufacturer? Buying more sellers' products would give a better perception of the variance between brands and the biological activity found and composition.

Reviewer 2 Report

The manuscript entitled “Antioxidant and antimicrobial properties of Helichrysum italicum (Roth) G. Don Hydrosol” by Bezek et al. reports the antioxidant and antimicrobial properties of hydrosol, which is generated as a by-product of the essential oil distillation process. I found the article interesting in terms of utilizing the by-product as a potential natural antimicrobial agent. However, there are several aspects that need to be improved in the manuscript.

1.     Manuscript requires extensive editing in terms of language and style.

2.     What is the rationale behind checking the anti-adhesive effect?

3.     Author has concluded that hydrosol has promising antimicrobial and antibiofilm activity against S. aureus, which does not agree with results. There is no activity noted against S. aureus.

4.     It should be discussed in the article why S. aureus is not susceptible to hydrosol. 

5. Discussion part must be improved, especially in case of antiadhesive properties.

Reviewer 3 Report

The article submitted for publication in Antibiotics describes the results obtained when essential oils, purchased by a company, of Helichrysum italicum and its derived hydrosol were tested for antimicrobial and cytotoxicity effects. The results are encouraged in the view of searching new active constituents needed to fight antibacterial resistance and the article is suitable for publication in this journal after the following revisions:

Line 34: before starting to explain the role of plant-derived metabolites in fighting antibiotic resistant, I suggest describing firstly the role of natural products and, in particular, both derived from terrestrial and marine organisms. At this proposal, the authors should cite this review on Antibiotics (Casertano, M.; Menna, M.; Imperatore, C. The Ascidian-Derived Metabolites with Antimicrobial Properties. Antibiotics 2020, 9, 510. https://doi.org/10.3390/antibiotics9080510).

In results’ section, the data should be presented in a table to give a clear overview of each experiment’s results. Moreover, some references are indicated in the main text as first author and year, please ensure that all references are showed in [].

The references should be adjusted. There are many mistakes about names and all journal names should be abbreviated.

Round 2

Reviewer 3 Report

The authors addressed all the raised comments.

The manuscript is suitable for publication in Antibiotics.